# Patients’ Buying Behavior for Non-Reimbursed Off-Loading Devices Used in Diabetic Foot Ulcer Treatment—An Observational Study during COVID-19 Pandemic from a Romanian Physical Therapy Unit

**DOI:** 10.3390/jcm12206516

**Published:** 2023-10-14

**Authors:** Iulia Iovanca Dragoi, Florina Georgeta Popescu, Frank L. Bowling, Cosmina Ioana Bondor, Mihai Ionac

**Affiliations:** 1Department of Vascular Surgery and Reconstructive Microsurgery, “Victor Babes” University of Medicine and Pharmacy, 2 Eftimie Murgu Square, 300041 Timisoara, Romania; iulia.dragoi@umft.ro (I.I.D.); frank.bowling@manchester.ac.uk (F.L.B.); mihai.ionac@umft.ro (M.I.); 2Discipline of Occupational Health, “Victor Babes” University of Medicine and Pharmacy, 2 Eftimie Murgu Square, 300041 Timisoara, Romania; 3Department of Surgery & Translational Medicine, Faculty of Medical and Human Sciences, University of Manchester, Oxford Rd., Manchester M13 9PL, UK; 4Department of Medical Informatics and Biostatistics, University of Medicine and Pharmacy “Iuliu Hațieganu”, 8 Victor Babeș, 400000 Cluj-Napoca, Romania; cbondor@umfcluj.ro

**Keywords:** diabetic foot ulcer, off-loading, diabetic peripheral neuropathy, COVID-19 pandemic, buying behavior, protective footwear, insoles

## Abstract

Diabetic foot ulcer non-reimbursed treatment depends on multiple factors, including the patient’s buying behaviors. Factors affecting buying behaviors for the removable off-loading devices are not completely understood. The aim of this study was to investigate the patients’ buying behaviors of the removable off-loading devices and their influence on the DFU treatment outcomes remotely monitored during the COVID-19 pandemic. In this prospective observational study, 45 patients affected by diabetic peripheral neuropathy, with/without peripheral arterial disease, with foot ulcers treated with removable devices were remotely monitored. Prefabricated removable cast walkers, insoles, and therapeutic footwear were the proposed off-loading methods. Patients affected by high blood pressure (*p* = 0.018), peripheral arterial disease (*p* = 0.029), past amputations (*p* = 0.018), and ulcer on the left foot (*p* = 0.007) bought removable cast walkers. Rural provenience (*p* = 0.011) and male (*p* = 0.034) did not buy a removable walker. The healing rate was 69.4%, while the median healing time was 20 weeks. High blood pressure negatively influenced healing time (*p* = 0.020). Patients who bought the most efficient treatment method for DFUs were females from urban provenience, with amputation in the past, with peripheral arterial disease, and with high blood pressure.

## 1. Introduction

Diabetes mellitus (DM) has reached pandemic proportions, affecting millions of people worldwide, the majority being diagnosed with type 2 diabetes [1]. In Romania, diabetes showed a prevalence of 11.6% in 2014 [2]. Diabetes mellitus and its micro and macro complications have become the fourth cause of death globally [3]. Diabetic peripheral neuropathy (DPN) is one of the complications affecting people with type 2 DM with different global percentages (23–70%) [4,5,6,7,8].

Peripheral arterial disease (PAD) has lately suffered an increasing tendency globally, with a higher increment in the prevalence of PAD in low- and mid-income countries [9].

Among all factors, smoking and DM are the most important risk factors for PAD [10,11].

Smoking increases with up to 3-fold the risk of PAD [12].

An Atherosclerosis Risk in Communities (ARIC) study reported an ≈4 relative risk (RR) of developing PAD for smokers (pack-years ≥ 25) when compared to never-smokers [13].

People affected by DM have a 1.7 risk of developing PAD [14], with DM being strongly associated with a severe form of PAD.

Adults with diabetes and hemoglobin A1c ≥7% have a 10.3 hazard ratio (HR) for chronic limb-threatening ischemia (CLTI) when compared with people not affected by DM. This explains the high rates (70%) of limb loss in this category of patients [15].

Epidemics of diabetes [16] and conventional smoking prevalence [17] contribute to constant changes in PAD prevalence and incidence worldwide.

High blood pressure (especially systolic) is another risk factor for PAD, and it was associated with CLTI [18].

Diabetic peripheral neuropathy and/or PAD can lead to diabetic foot syndrome. Late recognition of DPN might bring to late intervention, diabetic foot ulcer (DFU), and further amputation, increasing, therefore, the morbidity and mortality rate [19,20,21].

In patients with diabetes, up to 85% of amputations have a history of DFUs [22]. Patients with DM have a 20-fold risk of lower extremities amputations (LEAS) compared to people not affected by diabetes [23]. One Romanian study reported a prevalence of 14.85% of DFU history and a prevalence of 3.60% of LEAS history in people affected by DM [24]. After the first major amputation, the general mortality rate at one year is about 40% [25] and 80% at five years [26].

Aggressive foot care programs can lead to a reduction in amputation rates by 45% to 85% [27].

Diabetic foot-related wounds standards of care have been published with particular attention being attributed to off-loading [28].

Different off-loading solutions have been studied and analyzed for their efficacy in treating a DFU [29].

As an alternative to the Total Contact Cast (TCC), removable cast walkers (RCW), due to their efficacy in off-loading a DFU [30,31], gained interest and became a commonly used method [32,33].

In patients at risk of developing a diabetes-related foot ulcer, therapeutic footwear was considered for primary and secondary prevention [34,35,36,37].

A literature review reported the preventive effects of special therapeutic footwear [38], with no evidence for therapeutic footwear used for the treatment of active DFUs [28].

Rigid rocker-sole footwear [39], by reducing plantar pressures [40,41,42,43] and the mechanical load under the insensate foot, has demonstrated an effect on re-ulceration [44].

A strong correlation between the effectiveness of rigid rocker-sole footwear in preventing foot ulcers and the amount of wearing time has been demonstrated [45,46,47].

Insoles were also considered for their role in both prevention and reducing the recurrence of DFUs [48,49].

Diabetic foot suffered during the COVID-19 pandemic, and DFU management changed from in-clinic wound care to home-based self-care [50,51]. Diabetic foot care, including regular foot examination, dressings, and off-loading, was shifted toward patients’ self-management and responsibility remotely assisted by the clinicians [52,53].

Foot captures were sent by patients through mobile phone applications to their clinicians for treatment assistance [54,55,56].

Several studies reported different treatment protocols for DFUs during the pandemic [57,58,59,60,61,62].

In some situations, diabetic foot treatments were moved to non-reimbursed private care due to the reported disruptions in the public system [59].

When the treatment costs are supported by the patients, the outcomes can suffer more.

Off-loading device costs are primary determinants for the patient’s selection, irrespective of other preferences [63].

In the presence of self-supported costs for DFU care, there are few previously published papers that analyzed the factors that influenced patients’ choices for off-loading devices [64,65].

It has been recognized that there is a need for a more collaborative relationship with patients [66], which implies constant adaptation of the medical approach for the patient’s needs and treatment self-preferences. Patient preferences drive off-loading device decisions and acceptance, possibly increase adherence [67], and further influence healing.

The aim of this article was to present the patients’ buying behavior of non-reimbursed off-loading devices and its influence on the DFU treatment outcomes remotely monitored during the COVID-19 pandemic.

## 2. Materials and Methods

### 2.1. Study Design and Participants

This observational prospective cohort study implied the monitoring of patients from one private out-patient Physical Therapy Unit based in Timisoara, Romania, between March 2020 and February 2022.

All the patients aged over 18 years old who on admission had previous diagnostic of type 1 or type 2 DM with moderate to severe DPN, with/without PAD and with at least one active DFU (plantar/non-plantar), who were addressed for off-loading procedures, were consecutively included in the study. Diabetic-related neuropathy stage, PAD, and other DPN-related complications like unilateral/bilateral Charcot neuroarthropathy were previously diagnosed by the patient’s medical team. The two variables used, PAD and DPN, were taken from the patient’s medical files and were previously diagnosed by the medical team (vascular surgeons’ assessment and measurements), all being performed before the patients were sent for the off-loading treatment session. PAD was diagnosed using ankle and brachial index values (according to the Intersocietal PAD guideline (2023 update—https://iwgdfguidelines.org/wp-content/uploads/2023/07/IWGDF-2023-05-PAD-Guideline.pdf, accessed on 13 August 2023)), while DPN was diagnosed using Michigan Neuropathy Screening Instrument (MNSI), Neuropathy Disability Score (NDS), The Neurological Symptom Score (NSS), while DPN severity was diagnosed using the values in mV for vibration perception threshold (VPT) measured using the Neurothesiometer. Foot ulcers were graded according to the University of Texas Diabetic Foot Ulcer Classification [68] by the same medical team. Grade 1–2 A–D ulcers were included in the study. Grade 3D DFUs, addressed with proper local care and compatible with home assistance, were also included. Some patients had a recent history of endo/bypass-revascularization procedures and/or lower limb minor amputations.

Patients with grade 0 DFUs, severe wounds with infections/ischemia incompatible with home care (grade 3 D), foot wounds other than diabetic foot-related ulcers, under dialysis, severe cardiovascular disease (CVD), and severe retinopathy were excluded from the study. Severe foot and/or lower limb deformities incompatible with the available prefabricated off-loading devices and patients with major amputations were excluded from the study, as well as severe instability either due to diabetic neuropathy or other neurological-related conditions affecting gait. The presence of psychiatric conditions or any other psychologically related states, such as fear or just the unjustified rejection of wearing any off-loading devices, was followed by the exclusion from the study.

The monitoring period started with the first visit to the Physical Therapy Unit, where the off-loading treatment was applied.

Patients were considered lost from the study if during the monitoring any complications appeared (severe infections followed by hospitalization with/without major consecutive amputations, or acute ischemic events followed by hospitalization with/without major lower limb amputations).

All the patients included in the study sent for DFU off-loading were initially treated in one single session in the vascular surgery and reconstructive microsurgery department with DFU debridement, local wound care, and dressings.

At the first visit a podiatric assessment and off-loading procedures were performed. Properly trained physical therapists informed all patients about all foot care possibilities, available solutions, benefits, risks, and treatment costs.

All patients received written educational materials released by the Podiatry Association in Romania [69] on foot self-care, diabetic foot condition, and its possible complications. All the patients were informed about their data being processed and used in the study. All the patients signed written informed consent. The study was approved by the University of Medicine and Pharmacy “Victor Babeș” Timișoara Ethics Committee, released and registered under Nr. 50/21.09-14.10.2020.

Due to COVID-19 restrictions and the Physical Therapy Unit’s pandemic safety measures, after the first visit all patients had to adapt to home self-wound care. To avoid in-clinic re-calls, the off-loading solutions were reduced only to the available RCW [70].

The treatment methods choices for both affected and non-affected limbs are represented in Figure 1.

For the affected limb, the methods choices were RCW [71,72,73], therapeutic footwear [74] with inserted prefabricated mono-layer/multi-layer foot insoles [75,76], and personal off-the-shelf footwear with inserted prefabricated foot insoles.

For the non-affected limb, the method choices were therapeutic footwear with inserted prefabricated foot insoles or personal off-the-shelf footwear with inserted prefabricated foot insoles. For the situation when a DFU was bilaterally present, the method choices were bilateral RCW, RCW for the most affected and therapeutic footwear with inserted prefabricated foot insoles for the contralateral limb, and therapeutic footwear with inserted prefabricated foot insoles for both limbs.

All patients have been advised about the most suited off-loading treatment and the possibility to choose the preferred one or even reject all.

After the patients’ choices, off-loading treatment was applied, and all patients were remote-assisted. Patients’ feet were monitored by the same trained physical therapist through foot photography after detailed instruction was presented about the image’s quality needs. Foot photography was taken by the patient or by a member of the patient family and further sent to the physical therapist via a private end-to-end coded social mobile app [77].

The wound treatment indicated by the medical team was performed by the patient or with the family’s assistance, approx. 2–3 days each.

The foot photography implied captures of plantar, lateral, medial, and dorsal aspect of both feet. Any complicated cases identified from the pictures, were re-called for appropriate assessment and/or further sent for management under medical supervision.

At the end of the study (1 March 2022), each patient was re-contacted by phone and asked if any previous healed DFU reoccurred or if any new DFU to either foot was noticed by the patient or by the patient’s family.

Demographic and anthropometric data, smoking status, lifestyle, as well as diabetes-related data (HbA1c, etc.) and its complications (DPN stage, Charcot foot, etc.) were collected from the participant’s medical files. Patients were asked about their physical activity (PA) level. An active lifestyle was considered if they were still active at work or had at least 5000 steps daily; otherwise, they were considered sedentary. Other participants’ comorbidities (PAD, high blood pressure—HBP, cardiovascular disease—CVD, etc.) and the previous and present DFU treatment-related procedures were also extracted from the participant’s medical records. The number of past/present DFUs and the past/present DFU localization were recorded.

The present DFU localization was documented on separate anatomical foot areas, as seen in Figure 2.

Amputations number and type, foot deformities, and Charcot-specific deformities were documented.

### 2.2. DFU Dressing, Bandages, and Padding Procedures

Home-based local management of DFUs was maintained based on the medical team’s advice. The dressings used were chosen by the medical team (vascular surgeon before patients were sent for off-loading). In the case of wounds with excessive exudate, moist wound dressings were used with high moist absorption properties, while for less moist or dry wounds, gauze dressings were used. Available materials were used to prepare the foot for the RCW application. After the dressing application (Figure 3a), a cotton tubular bandage (Figure 3b) was used to protect the leg from direct contact with the padding material. Felt/medical wool was used for padding, as seen in Figure 3c [78]. To keep it in place, the felt/medical wool padding and an elastic tubular fixation bandage were applied (Figure 3d). Before the application of the RCW, a knee-high sock (Figure 3e) was used for bandages and paddings fixation and high protection.

### 2.3. Off-Loading Methods

#### 2.3.1. Removable Casts Walkers

When an RCW was used as an off-loading method, different heights and sole types were considered. According to the guidelines on off-loading foot ulcers in persons with diabetes [28] and the manufacturer recommendations [79], the height of the RCW was selected based on the localization of the DFU, presence of amputations, and ischemia/infection, while the RCW sole type was used based on the presence of the patient’s ability to maintain balance. All patients with unilateral/bilateral DFUs were considered for off-loading procedures. A night off-loading device was used for the heel wounds [80]. A detailed algorithm is represented in Table 1.

Table 1 represents the algorithm for the RCW types used for different clinical situations. 

The devices were manufactured either with a knee, ankle, or below ankle height, as seen in Figure 4a–c.

The height of the device was selected by the main physical therapist. Patients had the possibility to accept a knee-height device or to choose a reduced height device.

The RCW sole was either a full rocker (Figure 5a) or a flat (to-ground), more stable sole (Figure 5b).

A full rocker-bottom RCW was avoided in the presence of any condition affecting postural stability [81], any psycho-emotional reactions (fear of fall, etc.), or when postural instability during gait was observed while testing the device; thereafter, a device with a flat (to-ground), more stable sole was selected for safety reasons.

The differences between the impact of the different types of RCW used were not considered for this study.

##### RCW Internal Components Personalization

The device’s internal components were adapted by the main physical therapist using the manufacturer’s recommendations and the user manual instructions [82]. The DFU localization was imprinted on the RCW internal insole. Patients were not consulted for the RCW internal insole personalization, this procedure being clinician-dependent for appropriate DFU local off-loading.

For proper off-loading of only plantar regular-shaped wounds (Figure 6a), marking and imprinting the DFU on the removable device’s internal insole was required. This procedure was obtained by dressing the DFU using a transparent sterile dressing (Figure 6b) that allowed for the precise DFU location to be identified, marked (Figure 6c), and imprinted on the inner insole of the RCW. The patient was asked to walk for a few steps with the RCW in such a manner that the DFU contour during movement was imprinted on the mid-layer of the RCW insole (Figure 6d). A 45° cut using a blade was performed in the mid-layer and continued in the deep layer of the RCW insole until a cone shape was obtained, as seen in Figure 6e.

For non-plantar DFUs, irregular DFUs, and DFUs associated with skin grafts, respectively, the insole kit was used in its factory configuration.

The differences between the impact of the different RCW internal insoles personalization used were not considered for this study.

#### 2.3.2. Therapeutic Footwear with Inserted Prefabricated Foot Insoles

##### The Therapeutic Footwear

The therapeutic footwear (prefabricated medical-grade footwear) selection [74] was based on each medical situation, foot type, and presence of deformities and according to the guidelines on the prevention of foot ulcers in persons with diabetes [28].

Therapeutic footwear was proposed with the aim to reduce the risk of developing a pre-ulcerative lesion or a DFU in the contralateral foot during the treatment with RCW or as an alternative method for off-loading a DFU when the prescribed RCW was rejected/contraindicated or not tolerated by the patient.

The therapeutic footwear was either for outdoor or indoor usage (sandals or closed) and had different designs and colors. Therapeutic footwear presented with a high-density (rigid) fiberglass rocker mid-layer (Figure 7a), either with a forefoot rocker out-sole and a heel lift (Figure 7b) or a full rocker out-sole (Figure 7c). A full rocker out-sole was prescribed when enough ankle range of motion (ROM) was present, while a forefoot rocker out-sole with heel lift was prescribed when ankle ROM was less than 90° of dorsiflexion. In the situation of insufficient degrees of ankle plantarflexion, footwear with forefoot rocker out-sole and heel lift was avoided. Therapeutic footwear inner insole was removed and replaced with prefabricated foot insoles.

The footwear shape and material flexibility were chosen based on the patient’s foot anatomy and/or the presence of flexible/rigid deformities. Footwear modifications were considered in the presence of deformities incompatible with the shoe factory configurations (Figure 8) or when, after wearing trials, areas of skin redness or irritations were observed [83].

Patients had the possibility to receive one or multiple pairs of therapeutic footwear.

The main physical therapist recommended that at least one pair of outdoor usage footwear and one pair of indoor usage footwear should be acquired by all patients. Patients self-decided the number of pairs, design, color, etc., while the physical therapist decided the footwear size and shape based on clinical foot measurements and the presence of deformities.

The differences between the impact of the different types of footwear used were not considered for this study.

##### Prefabricated Foot Insoles

Based on the foot morphology, available prefabricated “off-the-shelf” insoles [28] either poron moderate arch high [76], as seen in Figure 9a, or high arch contoured ethyl vinyl acetate (EVA), either mono-layer or multi-layer [75], as seen in Figure 9b,c, were selected by the main physical therapist and used inside the therapeutic footwear with/without molding and wedging.

The differences between the impact of the different types of insoles used were not considered for this study.

#### 2.3.3. Personal off-the-Shelf Footwear with Inserted Prefabricated Foot Insoles

If the RCW or the therapeutic footwear were not accepted or tolerated, then prefabricated foot insoles were recommended to be introduced in the personal off-the-shelf footwear (prefabricated footwear that has not been modified and has no intended therapeutic functions) [28].

#### 2.3.4. Personal off-the-Shelf Footwear

Personal off-the-shelf footwear was not proposed as an alternative off-loading method for the affected limb, nor for preventive measures for the contralateral limb.

Personal off-the shelf footwear was kept for the non-affected limb if the above proposed therapeutic options were rejected or not tolerated by the patient.

### 2.4. Statistical Analysis

The patients’ group/subgroups were described with qualitative and quantitative variables. Absolut and relative frequencies, arithmetic means ± standard deviations, medians (25th–75th percentiles) were used as descriptive statistic parameters according to the distribution of the variables.

The qualitative characteristics of the two subgroups were compared using statistical tests: chi-square test/Fisher’s exact test. After using the Shapiro–Wilk test for normal distribution, the quantitative characteristics were compared with the t-test or with the Wilcoxon rank-sum test.

Time to event was analyzed with univariate and multivariate Cox proportional hazards regression analysis. The statistically significant variables in univariate analysis were introduced in multivariate analysis, enter model. Hazard ratio and its 95% confidence interval (CI) were presented.

A significant *p*-value was considered to be *p* < 0.05. Data were statistically analyzed with SPSS 25.00 [84].

## 3. Results

Forty-five patients met the inclusion and exclusion criteria and were included in the study.

### 3.1. Participants’ Characteristics

Participants’ characteristics are represented in Table 2. In the study group, patients were aged between 42 and 85 years old, more than two-thirds were male, 22 (48.9%) patients were overweighed, and 20 (44.4%) patients were obese. The diabetes duration was between 3 months and 38 years. Severe neuropathy was present in 42 (93.3%) of cases. Peripheral arterial disease was present in more than 70% of all patients, and approximately half of the patients presented CVD, with more than 80% presenting HBP.

### 3.2. Study Group Diabetic Foot Characteristics

After the podiatric evaluation, we found 40 (88.9%) patients with one DFU, 4 (8.9%) patients with two DFUs, and 1 (2.2%) patient with three DFUs (Table 3). Only two (4.4%) patients presented with bilateral DFUs. In more than two-thirds (66.7%) of patients, a history of DFUs was reported. History of revascularization was reported in 24 cases from all patients with PAD (72.7%), representing 53,3% of all cases.

There were 50 different DFUs in the study group, represented in Figure 10. The most common (64.4%) localization of DFUs was observed at the level of the metatarsal heads.

### 3.3. Study Group Treatment

A removable cast walker was indicated as the first treatment option for the 43 patients with unilateral DFUs. In 26 (60.5%) cases, an RCW was accepted, while 17 (39.5%) patients accepted the second treatment option (therapeutic footwear with inserted prefabricated foot insoles). For the affected limb, none of the patients were treated with personal off-the-shelf footwear with inserted prefabricated foot insoles, nor personal off-the-shelf footwear alone. Of the 26 patients with RCW, 22 (84.6%) of them chose one pair of therapeutic footwear with inserted prefabricated foot insoles for the not-affected limb, 3 (11.5%) chose two pairs of therapeutic footwear with inserted prefabricated foot insoles, and 1 (3.8%) chose personal off-the-shelf footwear with inserted prefabricated foot insoles.

Of the 17 patients without RCW who chose therapeutic footwear with inserted prefabricated foot insoles for the affected limb, 14 (82.4%) patients chose one pair of therapeutic footwear with inserted prefabricated foot insoles, and 3 (17.6%) chose two pairs of therapeutic footwear with inserted prefabricated foot insoles.

For the two patients with bilateral DFUs, the first treatment choice was a knee-height RCW on the most affected foot and an ankle-height RCW for the less affected foot. Both patients accepted an RCW on the most affected foot, while for the less affected limb, one patient chose one pair of therapeutic footwear with inserted prefabricated foot insoles, and one chose two pairs, respectively.

All patients chose prefabricated foot insoles.

Treatment costs were dependent on the type of the off-loading devices, while the assessment costs at initial podiatric evaluation had the same costs. Minimum total cost was 164 euro, the maximum total cost was 660 euro, while the average cost was 381 ± 107 euro. RCW had lower costs when compared to therapeutic footwear. The more devices were chosen by the patients, the more the treatment costs increased (r = 0.59, *p* < 0.001).

### 3.4. Contributing Factors for Off-Loading Choices and DFU Healing

Of the total of 28 patients who accepted an RCW for the ulcerated limbs, 4 (14.3%) were lost from the study. Of the remaining 24 patients with RCW, 18 (75%) were completely healed (15 with one DFU, 2 with two DFUs, and 1 with three DFUs). Complete healing was reached in less than 8 weeks for seven (27.8%) patients, between 9 and 16 weeks for nine (50%) patients, and between 17 and 52 weeks for the other four (22.2%) patients (Table 4). Both patients with bilateral DFUs were healed: one in 4 weeks and the other in 11 weeks. Those who were unhealed (six patients with one DFU) were followed in this study for at least 29 weeks.

Of the 17 patients who did not accept an RCW for the ulcerated limb, 5 (29.4%) patients were lost from the study. Of the remaining 12 patients, 7 (58.3%) patients were completely healed (6 with one DFU and 1 with two DFUs).

Of those who were healed, all seven reached complete healing in less than 8 weeks. Of the remained five (41.6%) unhealed patients, four patients had one DFU, and one person had two DFUs. Those who were unhealed were followed in this study for at least 19 weeks.

Of the total of 25 healed patients, 2 (8%) patients re-ulcerated after complete healing in 46 weeks of mean period follow-up.

In Table 4, a comparison between patients with and without a removable walker was represented. Patients who significantly more accepted the treatment with an RCW were women, from urban provenience, with bilateral DFUs, and with history of amputation.

A statistically significant higher number of patients affected by PAD accepted an RCW. Of all patients affected by PAD, those who benefit from revascularization (18 (78.3%)) were more prone to accept RCW than those who did not benefit from revascularization (4 (50%)), without being statistically significant. Of those affected by PAD and a history of amputation, 19 (76.0%) accepted RCW when compared to 4 (50%) with PAD and without amputation.

Of the total number of patients affected by HBP, 23 (71.9%) patients accepted an RCW, while from the total number of patients not affected by HBP, only one patient (16.7%) accepted an RCW (*p* = 0.018).

A total of 13 patients out of 18 (72.2%) with rural provenience were aged less than 65 years compared to only 11 out of 26 (42.3%) patients from urban areas (*p* = 0.05). In the rural group, half of the patients had a previous history of DFUs, in comparison with 80% from urban areas (*p* = 0.031). In the rural group, 7 (38.9%) patients had a previous history of amputations when compared to 20 (76.9%) patients from the urban area (*p* = 0.035). A total of 7 patients (38.9%) from rural areas had foot deformities when compared to 19 (73.1%) patients from the urban area (*p* = 0.023), with varus blunt in 7 (26.9%) patients from the urban area (*p* = 0.031). Of 18 patients with rural provenience, 7 (38.9%) accepted an RCW, while 20 (76.9%) patients from urban areas accepted an RCW (*p* = 0.011) (Table 4).

When the DFU was localized on the right foot, patients (8 (38.1%)) were less prone to choose an RCW when compared to patients who had a DFU localized on the left foot (18 (81.8%)) (*p* = 0.003) (Table 4), maybe due to the active lifestyle and the rural area of provenience. When comparing patients with DFUs on the right foot and with an active lifestyle with all other (sedentary or with DFUs on the left foot), of 8 patients who had a DFU on the right foot and with an active lifestyle, 2 (25%) patients chose RCW, while of all the other 35 patients, 24 (68.6%) chose RCW (*p* = 0.042) (patients with DFUs on both sides were not considered in this statistics); also, patients with DFUs on the right foot and with active lifestyle were preponderant from rural area (6 patients (75%) out of 8 patients), while of the other 35 patients, 11 (32.4%) were from rural areas (*p* = 0.045) (patients with DFUs on both side were not considered in this statistics).

In Table 5, the univariate and multivariate survival Cox regression is presented. The considered event was DFU healing. All collected factors affecting time to healing were analyzed first in univariate regression, while in multivariate regression, only significant factors were entered.

Of those patients who were included in the study, 55.6% patients healed. Of the 36 patients that remained until the end of the study, 25 (69.4%) patients completely healed.

Median survival time until complete healing for the total study group was 20 weeks (95% CI 7.15;32.85). For those who accepted an RCW median survival time until complete healing was 16 weeks (95% CI 0.44;31.56), while for those who did not accept an RCW median survival time until complete healing was 20 weeks (95% CI 8.64;31.36), *p* = 0.262.

Age significantly influenced time to DFU healing. Younger patients had a slower DFU healing time. Age remained significant in multivariate analysis.

Patients with HBP had a significantly longer DFU healing time. HBP remained significant in multivariate analysis. Patients with HBP had a 3.44 higher hazard ratio for non-healing DFUs when compared with patients without HBP.

Bilateral DFUs had five times longer DFU healing time when compared to unilateral DFUs; however, in the multivariate analysis, it did not remain statistically significant.

When HbA1c and smoking were analyzed in univariate analysis, no statistically significance was found.

When PAD was analyzed in univariate analysis, no statistical significance was found, while HBP results were statistically significant.

### 3.5. Encountered Situations during Remote Monitorization

The encountered situations during follow-up are described below.

#### 3.5.1. Remote Communication-Related Situations

The quality of the communication between the main physical therapist and the patient/patient’s family was influenced either by technology usage (device type, camera resolution, capture quality, etc.) or by the lack of patient/patient’s family technical skills in capturing relevant images (improper focalization, wrong framing of area of interest), as seen in Figure 11a,b. Patients living alone were more prone to capture improper photos either from the inability to use technical devices, physical inability to capture their own foot, improper room lightening level, poor eyesight, etc.

Patients living alone who were unable to self-capture their foot were instructed to use a mirror, as seen in Figure 12.

#### 3.5.2. Encountered Self-Treatment Errors

While analyzing the serial photos received from the patient/patient’s family, as expected, the DFU healing course and eventual onset of DFU complications were appreciated.

Foot photography also informed about the quality of DFU local care and revealed some errors related to dressings/bandage application, RCW usage, poor hygiene, adverse effects related to dressings and off-loading devices (RCW, footwear, insoles), any deterioration of the devices, DFU complications (infection, etc.).

Dressing-related errors were mainly due to patients’ inability to respect the medical team’s indications and instructions on the dressings and bandage characteristics. Improper choice of the dressing type and size, as seen in Figure 13a; insufficient surface coverage of the dressing, as seen in Figure 13b; wrong application technique of bandage, as seen in Figure 13c; and wrong choice of compression bandage type and wrong application technique, as seen in Figure 13d, were the main encountered errors.

Such situations were followed by corrective measures. Initial instructions for the patient/patient family were reminded.

#### 3.5.3. Encountered Complications

Complications seen on foot captures were related to intolerance to medical devices materials, infections, off-loading medical devices-related adverse effects.

Skin irritations (as seen in Figure 14) were encountered and were mainly due to allergies to dressings and padding materials used. Allergies were seen in the cases of patients who did not respect the dressings indications and used other available non-specific dressings. One particular allergy has been seen in the case of using the padding material directly on the skin due to the unavailability of a cotton tubular bandage that protects the direct contact of wool padding with the skin. Another particular irritation was seen in a patient who used a gauze dressing with adherent adhesive on the skin and not a semi-elastic gauze bandage, as per the medical team’s recommendations.

Dermatological treatment was requested and corrective measures were applied, or cessation of using the not-tolerated products.

Infections were suspected from the foot captures, as seen in Figure 15, and were followed by an urgent medical approach. Any sign of swollen, red, hot foot or any signs of DFU infected aspect were considered for an urgent medical approach.

Other complications seen on captures were derived from the usage of off-loading devices (footwear, RCW, insoles). Footwear-related skin irritations, as seen in Figure 16a, and RCW-related skin irritations with associated compression-related local ischemia, as seen in Figure 16b (possibly due to the incapacity to properly inflate the air camera of the device), were the main encountered situations.

Other possible secondary DFUs were related to either insoles of self-made improvisations, as seen in Figure 17a, or to the wrong fitting of the RCWs, as seen in Figure 17b.

Such complications were followed by reminding the correct instructions. Cessation of wearing the device until complete healing (in case of compression-induced ischemic wounds) and dressing and dermatological treatment in case of skin irritations were also considered.

#### 3.5.4. Poor Hygiene

Captures revealed situations of both foot poor hygiene and home ambiental poor hygiene. These situations were followed by clear instructions reminding of the necessity of proper foot basic care and clean ambient, especially where the dressing procedure took place.

#### 3.5.5. Medical Devices Deterioration

Captures also revealed deterioration signs of the medical devices used (Figure 18a,b), which were followed by the replacement of the device.

### 3.6. Patients Lost from the Study

At the end of the follow-up, nine patients were lost from the study. Of the nine patients lost from the study, six patients (33.3%) were from the rural area, significantly more than the three (11.1%) patients from the urban area (*p* = 0.048). The mean age of the patients lost from the study was 55.89 ± 9.80, significantly lower than the age of those not lost from the study, 63.42 ± 8.68 (*p* = 0.028). Four (44.4%) patients lost from the study were active with a DFU on the right foot, compared to those four (11.4%) active patients with a DFU on the right foot who were not lost from the study (*p* = 0.042). Four patients (44.4%) chose RCW for the affected limb, while five (55.6%) patients only chose therapeutic footwear for the affected limb. Eight (88.9%) patients chose one pair of therapeutic footwear with inserted prefabricated foot insoles for the non-affected limb, while one (11.1%) patient did not choose any therapeutic shoes for the non-affected limb. All nine patients lost from the study agreed with prefabricated foot insoles. None of the treatment options chosen by the patients lost from the study was significantly different from the treatment options chosen by the patients that were not lost from the study. Patients lost from the study had a mean monitoring period of 68.43 ± 16.54 weeks, approximately three times higher than the average monitoring period of those not lost from the study (23.72 ± 22.5), with no statistical significance (*p* = 0.267).

## 4. Discussion

The aim of the study was reached. All patients have been informed about RCW as the best treatment method for off-loading the DFU while in home care. The majority of patients (62.2%) chose RCW for the affected limb, while the rest of them chose therapeutic footwear with inserted prefabricated foot insoles. There were only a few studies from our knowledge analyzing what determines a patient to accept the off-loading proposed treatment solutions and what are the relations between patients’ choices and the severity of the condition.

### 4.1. Treatment Choices

In our study, RCWs were considered the first choice of treatment to off-load a DFU due to pandemic restrictions and reduced accessibility to medical care while patients were remotely monitored. Total contact casts (TCCs) are considered the first choice for off-loading a DFU [85,86]. The application of a TCC restricts access to wound care [70] and requires patient presence at the clinic, practitioners’ time, and skills, probably explaining the limited use of TCC among practitioners [33,87]. When TCC is not applicable, as in our study during the pandemic, non-removable walkers and RCW have also been proposed and appeared in the guidelines for DFU off-loading [28]. The removability of the selected devices enables patients/patients’ families to access the wounds.

In our study group, we observed that in severe cases, RCW was bought compared with the other proposed choices. A different but similar concept of adherence was analyzed in other studies, and it was found that the severity of neuropathy and DFU was associated with a better adherence to off-loading treatment [88].

In our study, therapeutic footwear with insoles was the second option of treatment for the affected limb when an RCW was not chosen by the patient or an RCW was contraindicated. Therapeutic footwear showed a tendency to decrease DFU recurrence in participants with a history of DFUs [40,44,45,49].

Therapeutic footwear was also proposed for the non-affected limb for preventive measures. In participants who had no previous DFUs, therapeutic footwear could prevent a foot ulcer [89]. Studies showed that patients with a history of plantar DFUs showed greater plantar pressures than patients without a history of ulceration [90]; thereafter, an “at risk” foot needs protection from abnormal mechanical load [91]. To off-load a “foot at risk” (IWGDF second and third risk class), special therapeutic footwear has been recommended for both indoor and outdoor use [92]. When using an RCW, due to the induced functional leg length discrepancy (LLD) and the increased pressures on the contralateral foot [93], secondary possible adverse reactions might appear [94] on the non-immobilized foot [95].

Only rigid rocker-sole footwear with off-loading properties was proposed in our study. When patients with a history of DFUs used therapeutic footwear with a rigid rocker sole, a reduced risk of plantar ulcer recurrence was observed [47].

In our study, patients benefit from one/two pairs of low-density EVA/poron prefabricated foot insoles for each pair of therapeutic footwear provided or for personal off-the-shelf footwear. Insoles’ protective effect has been studied, and their potential role in reducing the risk for ulceration/re-ulceration has been assessed [48,49].

### 4.2. Factors That Influenced Buying Behavior

Factors that influenced patients’ buying behavior were analyzed.

In our study, the frequency of patients who bought an RCW was higher in females, in urban provenience, in the case of past higher number of amputations, and in the presence of HBP and PAD. All patients with PAD were affected by HBP, too. The history of DFUs and amputations, which were previously not treated with any off-loading methods, influenced the buying behavior, too. The severity of the condition could explain the patient’s decision to take more aggressive actions to further prevent future complications. When adherence was studied, longer diabetes duration, males, absence of PAD, and the weight of the RCW were factors associated with the reduced adherence level [96].

The severity of neuropathy and DFU status was correlated with better adherence to off-loading devices [88,97,98].

In our study, males bought an RCW less than females. We are not aware of what determined this difference, but we can appreciate that males were more involved in activities of daily living (ADL) and professional activities that could have been negatively influenced by wearing an RCW.

Patients with rural provenience bought less RCW when compared to those with urban provenience. This could be explained by their younger age (being possibly more active) and reduced severity of their disease condition (less history of DFUs, fewer deformities, less history of amputations, and fewer varus blunt deformities, respectively). Patients with a DFU on the right foot, with an active lifestyle, with rural provenience, and less severity of the condition did not choose an RCW. We did not analyze this in particular, but we can hypothesize that those with a DFU on the right foot, being more active, did not choose an RCW for driving safety reasons. To access their nearest specialist centers, driving is a necessity, mainly for patients from rural areas that do not have other convenient transport options. Similar situations with regard to transport for patients living in rural areas have been reported in the literature [99,100].

In our study group, from all patients who previously had DFUs, only two patients benefited from off-loading therapy, possibly highlighting that off-loading procedures are not a commonly used method in the Romania DFU treatment guidelines, either by the lack of knowledge of standards of care in diabetic foot and/or the lack of skills in off-loading methods for DFUs, respectively, or by the podiatry profession missing in Romania.

In our study, the treatment cost also influenced buying behavior. Treatment costs were between a minimum of EUR 164 and 660. Patients who bought an RCW had significantly higher costs than patients who did not buy due to the cumulative costs of all the acquired devices or due to increased costs of some acquired devices. We found that costs increased with the number of chosen devices. The economic burden of hospitalization costs for patients with DFUs was previously reported [101,102]. The costs reported in Romania for DFU hospitalization were found to be 40% higher when compared to those without DFUs [103], still not so high as the costs reported in most of the European countries [104]. To our knowledge, there is no previously published paper analyzing the costs for DFUs outside hospitalization in Romania. Also, the previously analyzed costs of hospitalized DFUs did not include off-loading treatment, postulating that under treatment or lack of appropriate therapies for DFUs in patients with DM are missing as standard care in Romania [103].

Several studies identified that the purchasing costs for off-loading devices represented a barrier to achieving the proposed treatment [67], especially when the income was fixed and insufficient.

When compared to other countries’ reports, our study shows that off-loading costs for out-patients with a DFU are lower. Considering the average monthly net income for active workers (804 euro) [105] and the average monthly pension in Romania (375 euro) [106], the DFU off-loading costs still represent a high cost (47.4% of monthly net income and 101.6% from monthly net pension) for DFU off-loading treatment.

Conventional follow-up was two times more expensive when compared to tele-medical intervention in DFUs [107].

Increased cumulative costs for buying off-loading devices can also explain the buying behavior of the patients with rural provenience due to possible lower affordability. Implementing a strategy to have off-loading devices reimbursed by national health insurance could improve buying affordability.

### 4.3. Treatment Efficiency

Our study reports the self-home-treatment outcomes of DFUs managed with non-reimbursed removable devices during the COVID-19 pandemic. In our study, of the total number of patients initially included, 55.6% were completely healed. Of the total number of patients that remained until the end of the study, 25 (69.4%) patients completely healed.

In our study, we failed to demonstrate that the treatment of DFUs was more efficient with RCW and least effective with currently available therapeutic shoes.

In other studies, the healing rate was reported to be between 58% and 95%, respectively, and 81% when off-loading devices were used [108,109].

Diabetic Foot Disease management has been redefined during COVID-19, shifting DFU treatment to patients’ self-responsibility [59]. Even though remote management of DFUs has not been extensively utilized [56], its effectiveness in amputation prevention in people with DFUs has been demonstrated [52,55].

Recent studies that analyzed the outcomes of DFU treatment using home-podiatry assistance, physician-to-patient virtual consultations, and physician-to-home care telemedicine marked the new models of care for DFUs [51].

Studies analyzed the effect of the pandemic on DFUs, showing a likely beneficial effect of the lockdown period, with 72% of patients with a DFU remaining stable or having improved [110].

In previous studies, the effectiveness in the healing rates of a DFU treated with removable devices was compared with the effectiveness of TCC [111]. Removable cast walkers were associated with poor outcomes [30,112] mainly due to their removability and secondary poor patient adherence to wearing the device [97].

In our study, the median survival to complete healing calculated for the entire group was 20 weeks with a 95% CI (7.15;32.85).

In our study, we failed to demonstrate that the treatment of DFUs was more efficient with RCW and least effective with currently available therapeutic shoes. The median survival to complete healing for those who bought an RCW (16 weeks) was smaller than for those who did not buy an RCW (20 weeks) but did not differ significantly.

In other studies, the time to healing was reported to be between 35 and 85 days when off-loading devices were used [109]. The amputation rate and healing time were not negatively influenced by telemedicine intervention when compared to hospital standard care for DFUs [107].

When compared with other studies results on the healing time when an RCW was used to treat a University of Texas grade 1A DFU (58.0 +/− 15.2 days) [113], our study shows some differences, with a slightly longer healing period that could be explained by the complexity and different grades of the DFU in our case.

In our study, we identified a delay in median of 8 (3.25, 14.5) weeks from DFU onset to identification. An average delay of 15,3 days between DFU occurrence and DFU identification was previously reported [114].

The time to complete healing was age influenced, with younger patients having longer healing times. This could be explained by the activity level being higher in younger patients. Another possible factor explaining longer healing time in younger patients could be the reduced number of past amputations/DFUs, further explaining the unawareness of the possible complications associated with not accepting/respecting the recommended off-loading treatment.

In our study, HBP was shown to negatively influence healing time. HBP was found to be a negative prognostic factor along site to blood glucose level, DFU size, and presence of infection [115].

Bilateral DFU versus unilateral DFU had negatively influenced the healing time.

When age, HBP, and bilateral versus unilateral DFUs were analyzed in multivariate analysis, only HBP and age remained significant.

Two patients from the total group that accepted RCW re-ulcerated during the monitoring period. We appreciate these rates as being optimistic. The similarities with the data from the literature are possibly explained by the reduced activity level during the pandemic and especially the lockdown period. Studies reported high recurrence rates, with almost 50% of wounds to the contralateral foot and only 17% of past DFUs recurring to the same anatomical location [116].

In our study, some aspects related to the deterioration of the RCW, therapeutic footwear, and insole components were also reported during the follow-up. It was stated that insoles and therapeutic footwear mechanical properties could be impaired after 6 months of usage [86,117].

When special therapeutic footwear with off-loading properties was used, a suitable efficacy was proved in reducing DFU incidence. Special attention should be paid to the wearing time of therapeutic footwear as the effectiveness could be altered by the gradual decrease in the protective properties. There are still no available guidelines on the maximum utilization time and the need for renewal of protective footwear [38].

In our study, severe situations like infections and ischemic wounds related to RCW compression were identified through foot captures and referred for urgent medical approach.

Several other complications have been recognized during the follow-up. Foot complications during remote monitoring appear and need attention. Two patients from the total group, one with RCW and one without RCW, suffered infection-related amputations during follow-up. The removability of the devices used allowed for foot self-inspection and appreciation of any secondary wounds or infections. Medical device-related complications and any other complications were registered and approached accordingly. Other studies found that patients affected by neuropathy have difficulties while using removable cast walkers equipped with inflating airbags [63].

Adhering to self-care precautions related to off-loading has been recommended in previous studies to avoid complications and hospital attendance [53].

Poor self-care behaviors in persons affected by diabetes could relate to emotional distress in the presence of a DFU [118,119] and depression, further influencing the risk of DFU occurrence [120] and healing delay [121,122].

While analyzing the captures, poor self-care behaviors were noted. Poor foot hygiene, poor ambient hygiene, and delays in dressing sessions were the main encountered situations possibly related to emotional distress and incapacity to self-complete ordinary personal needs and daily common activities.

### 4.4. Study Limitations

The small sample size and not being a randomized, double-blind control trial are some of the major limitations of this study.

DFU classification in our study was not considered for separate analysis.

The limited available choice for the removable off-loading devices limited the treatment options for DFU care.

Another study limitation was not using questionnaires able to explain why the patients chose one treatment method over another.

Emotional aspects that could have possibly influenced the acceptance have not been analyzed.

Another limitation of our study was the inability to objectively measure adherence to the accepted off-loading devices and the level and type of physical activity during activity, daily living, and professional work when using the prescribed devices.

At the limit of the validity, 20% of patients were lost from the study.

Future research should consider the education level, patients’ posture, activity time and type during the day, and professional work for improving the off-loading plan, acceptance, and adherence.

Future studies should analyze the outcomes differences in self-independence patients and those who benefit from family support.

## 5. Conclusions

Patients’ buying behavior for off-loading devices was found to be dependent on medical condition severity and personal factors.

A severe condition (with HBP, increased number of amputations in the past, with PAD) determines whether patients buy RCW instead of therapeutic footwear.

Other factors that negatively influenced patients’ buying behavior for RCWs were rural origin, poor financial status, and male sex.

During the COVID-19 pandemic, patients with a DFU had a delay in addressing off-loading treatment.

Remote monitoring assisted by trained practitioners could be a method for self-independent patients with DFUs treated with removable devices.

Remote DFU treatment needs to be considered with precautions due to some possible encountered situations that could alter the outcomes (poor hygiene, bandage utilization errors, reduced capacity to use communication devices and capturing foot pictures, lack of specific consumables, lack of skills, and inability to properly self-care, especially when living alone, etc.).

### Recommendations

When foot captures are used to remotely monitor DFUs, improving the quality of pictures should be considered.

The deterioration of the RCW and insole components reported during the follow-up reminds us of the necessity to closely monitor the off-loading device status in order to prevent adverse events or poor treatment quality.

## Figures and Tables

**Figure 1 jcm-12-06516-f001:**
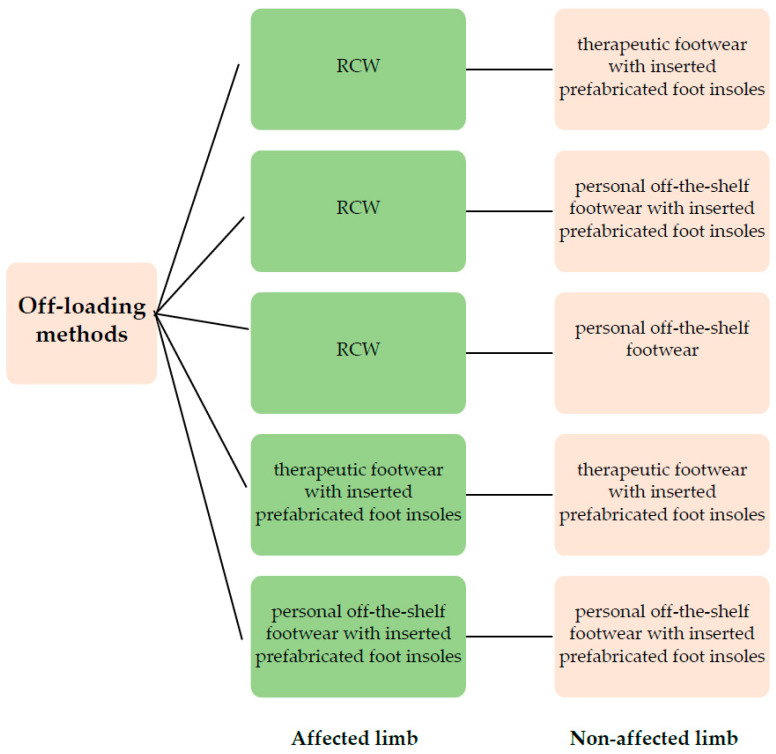
Treatment methods choices for both affected and non-affected limb.

**Figure 2 jcm-12-06516-f002:**
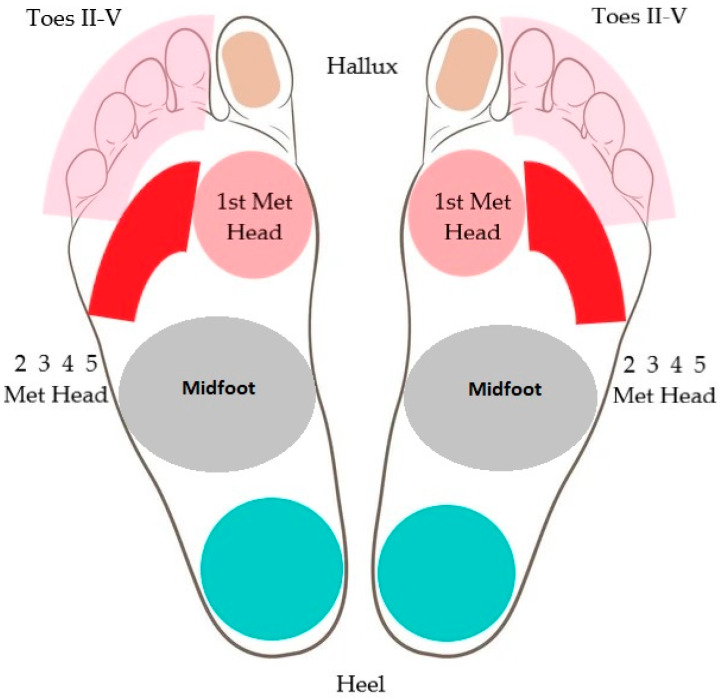
Foot anatomical areas represented as a chart for present DFU localization.

**Figure 3 jcm-12-06516-f003:**
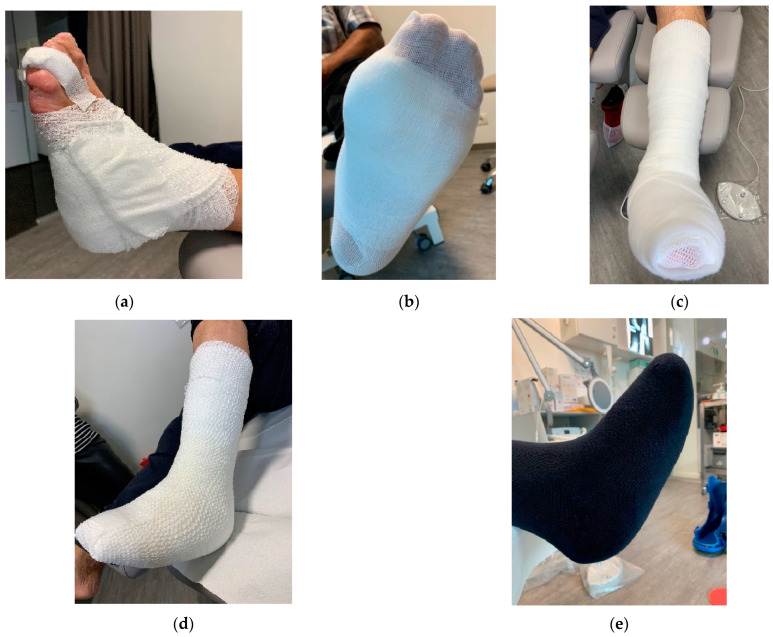
DFU preparation before applying the RCW: (**a**) dressing fixation technique using semi-elastic bandages with generous coverage; (**b**) cotton tubular bandage application avoiding direct contact with the padding material; (**c**) felt/medical wool application for protection from excessive pressure and contact with RCW; (**d**) stabilization of felt/medical wool padding using an elastic tubular bandage; (**e**) external knee-high sock used under the RCW.

**Figure 4 jcm-12-06516-f004:**
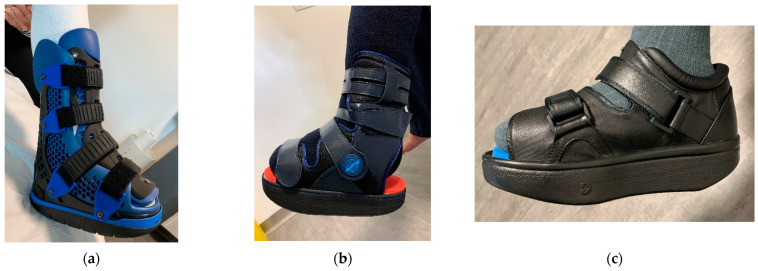
RCW types: (**a**) below knee height; (**b**) ankle height, either with a closed heel or a cut-off in the heel area; (**c**) below ankle height.

**Figure 5 jcm-12-06516-f005:**
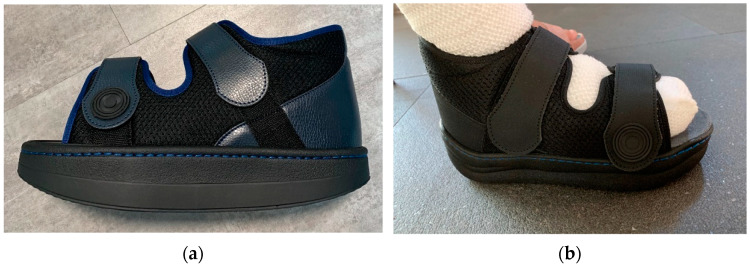
Types of RCW out-soles: (**a**) full rocker sole; (**b**) flat sole.

**Figure 6 jcm-12-06516-f006:**
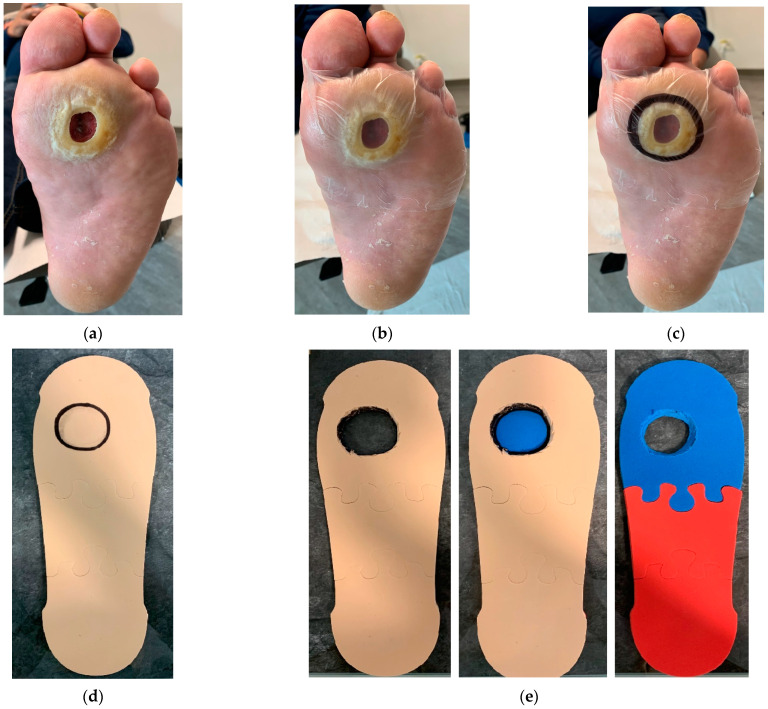
Personalization of the RCW insole: (**a**) DFU localization; (**b**) application of transparent sterile dressing; (**c**) DFU is marked on the transparent sterile dressing; (**d**) imprinting of the DFU contour on the mid-layer of the RCW insole; (**e**) a cone-shaped cut-off is performed in the mid and deep layer of the RCW insole to off-load the DFU area.

**Figure 7 jcm-12-06516-f007:**
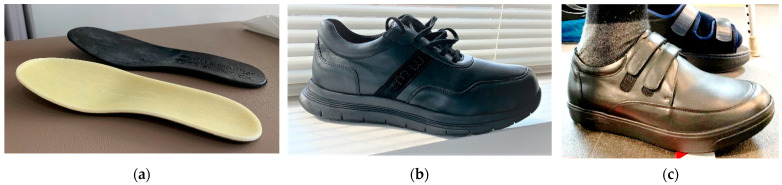
Therapeutic footwear sole components and characteristics: (**a**) high-density (rigid) fiberglass full rocker mid-layer; (**b**) forefoot rocker out-sole and incorporated heel lift; (**c**) full rocker out-sole without an incorporated heel lift.

**Figure 8 jcm-12-06516-f008:**
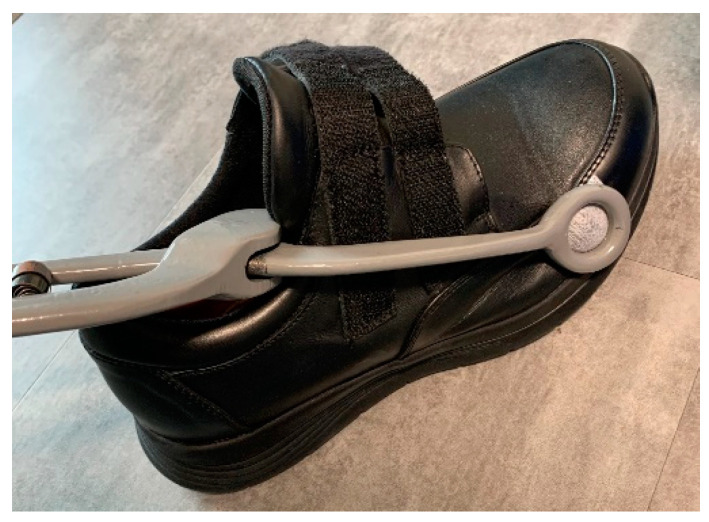
Representation of therapeutic footwear modification considered in the presence of foot deformities that do not accommodate with the shoe factory configuration.

**Figure 9 jcm-12-06516-f009:**
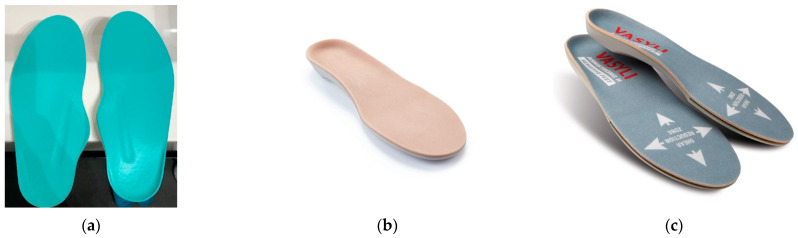
Prefabricated “off-the-shelf” insoles used: (**a**) poron moderate arch high foot insoles; (**b**) EVA contoured mono-layer foot insole; (**c**) EVA contoured multi-layer foot insole.

**Figure 10 jcm-12-06516-f010:**
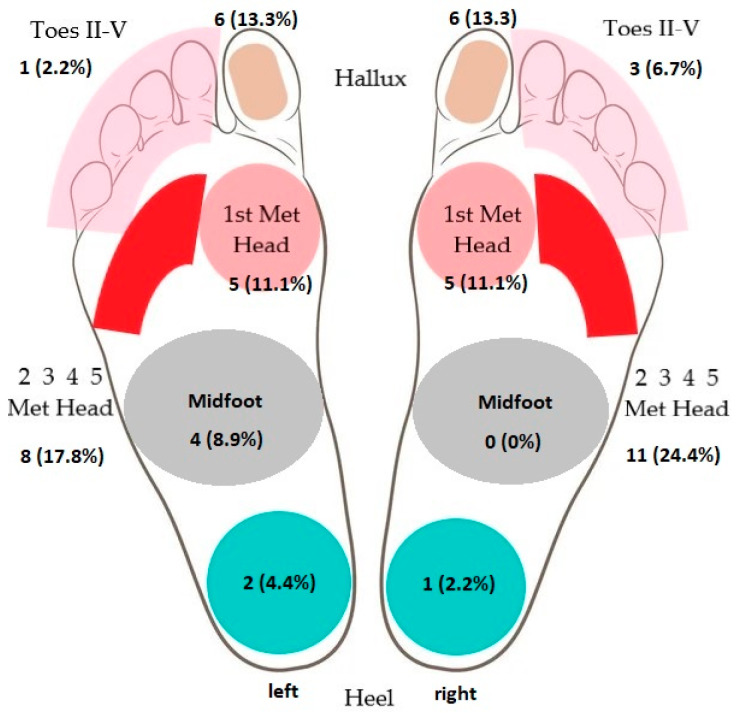
DFU localization on different foot anatomical areas.

**Figure 11 jcm-12-06516-f011:**
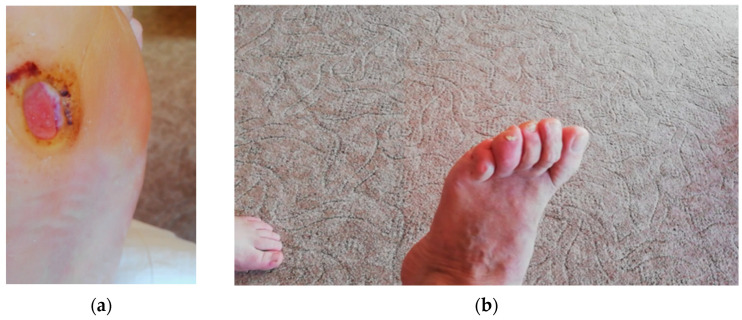
A representation of irrelevant foot captures: (**a**) improper focalization of the DFU; **(b**) wrong framing of the area of interest.

**Figure 12 jcm-12-06516-f012:**
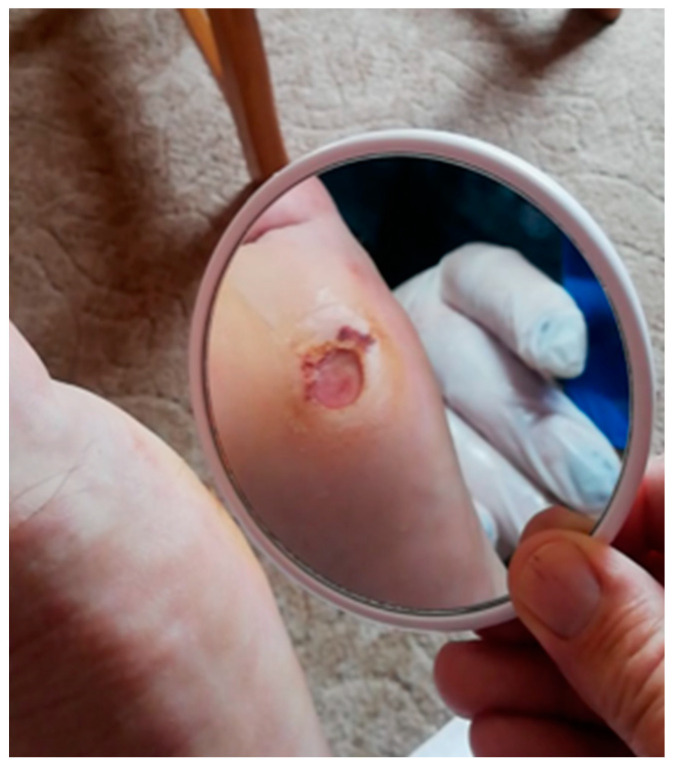
A representation of foot self-capture using a mirror in the case of a patient unable to approach his own foot.

**Figure 13 jcm-12-06516-f013:**
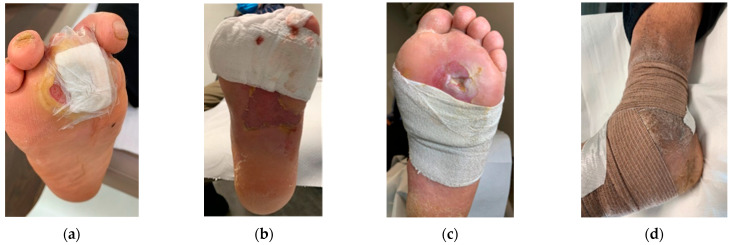
Representation of dressing-related errors seen on foot captures: (**a**) inappropriate appreciation of dressing (wrong size and type of dressing), resulting in displacement and insufficient coverage of the wound; (**b**) inappropriate choice of dressing size, resulting in insufficient coverage of the wound; (**c**) improper application technique, resulting in displacement of the dressing; (**d**) inappropriate bandage application technique with excessive compression, resulting in skin irritation and associated local edema.

**Figure 14 jcm-12-06516-f014:**
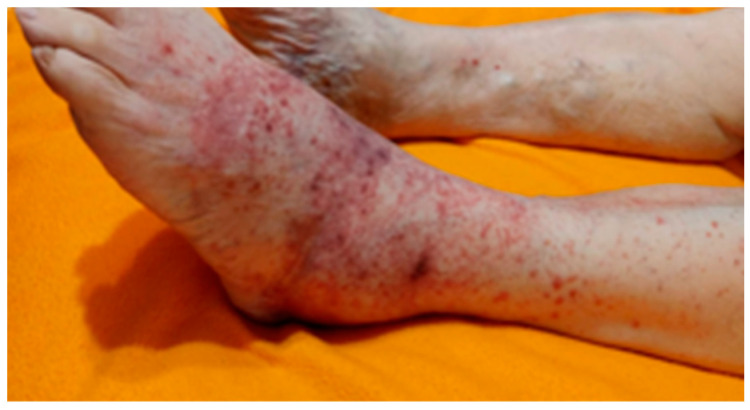
Representation of skin allergy to padding material seen on foot capture in a case where the protective cotton tubular bandage was not applied prior to the felt usage.

**Figure 15 jcm-12-06516-f015:**
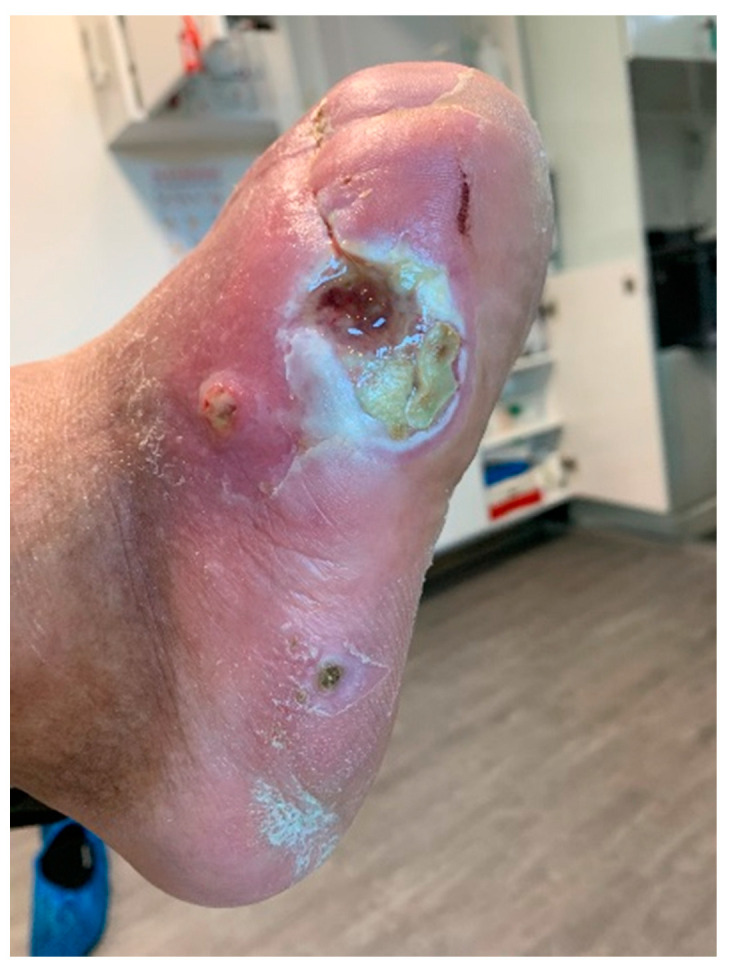
Capture of a swollen, red colored foot, with lateral abscess, showing signs of infection.

**Figure 16 jcm-12-06516-f016:**
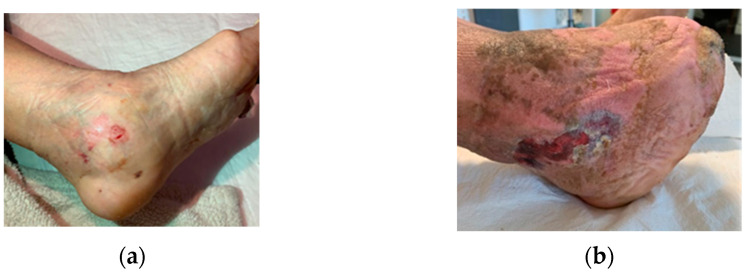
Representation of adverse effects associated with the usage of off-loading devices as seen on self-captures: (**a**) secondary wound on the medial malleolar area induced by direct contact with footwear; (**b**) compression-related local ischemia and skin irritation due to excessive tightening of RCW straps and excessive inflation of the air camera of the RCW.

**Figure 17 jcm-12-06516-f017:**
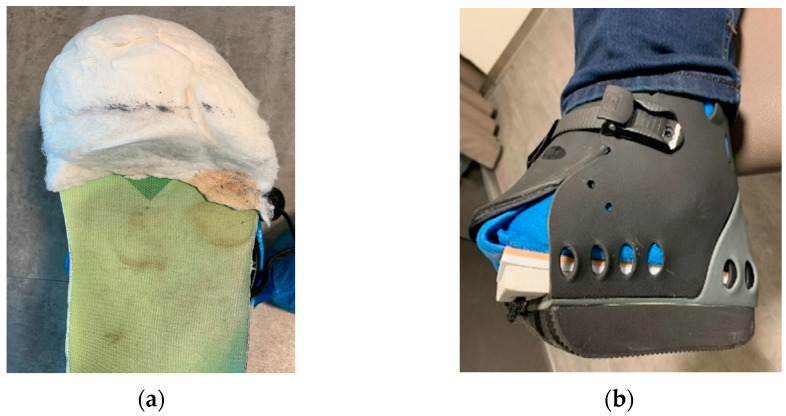
Representation of wrong/inappropriate usage of the off-loading devices as seen on self-captures: (**a**) self-improvisation of the insole that could explain the presence of blood associated with suspicion of secondary DFU; (**b**) inappropriate RCW application due to the inability to follow the provided instructions.

**Figure 18 jcm-12-06516-f018:**
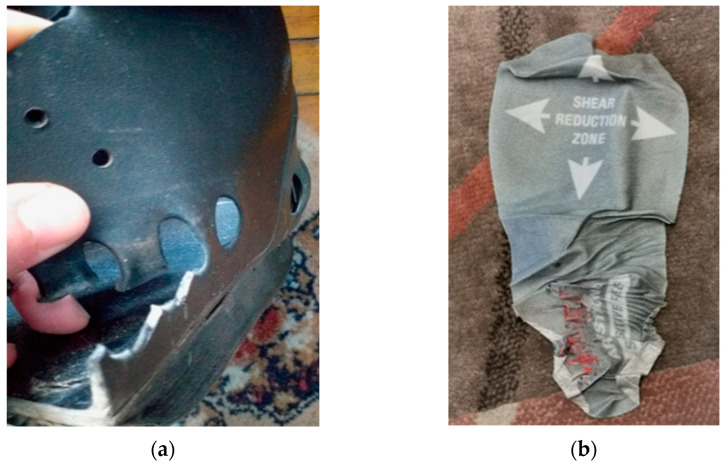
Representation of signs of deterioration of the medical devices components: (**a**) External semi-rigid material rupture due to varus knee with excessive load on the lateral side of the device; (**b**) Insole top-cover detached from the insole due to excessive moisture and inappropriate device cleaning.

**Table 1 jcm-12-06516-t001:** The proposed off-loading algorithm is based on the wound localization, level of amputation, and presence of PAD/infection and/or ischemia.

Proposed Treatment Algorithm	Condition Severity	DFU Localization	Amputation Level	Proposed RCW
1st choice	Unilateral DFU/unilateral DFU + mild/moderate infection/± mild/moderate ischemia), acute/chronic Charcot	Plantar/non-plantar 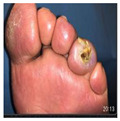 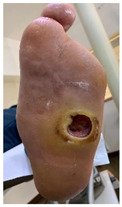	Toes, transmetatarsals, rays 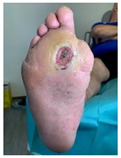 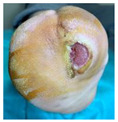	Removable knee-high off-loading device 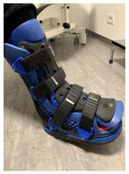 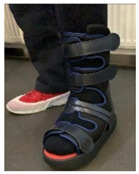
2nd choice	Unilateral DFU/unilateral DFU + mild/moderate infection/± mild/moderate ischemia), chronic Charcot	Plantar/non-plantar	Toes, transmetatarsals, rays	Removable ankle-height/below-ankle-height off-loading device
		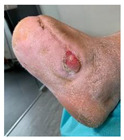 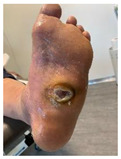	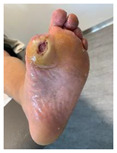 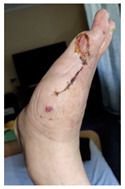	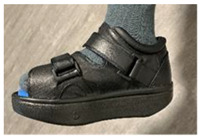
3rd choice	Unilateral DFU/unilateral DFU + mild/moderate infection/± mild/moderate ischemia), chronic Charcot	Plantar/non-plantar	Toes, transmetatarsals	Rigid fiberglass rocker-sole footwear + prefabricated foot orthotic
		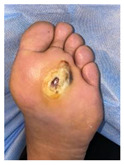	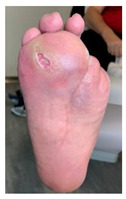	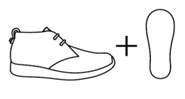
Other choices	Unilateral DFU/unilateral DFU + mild/moderate infection/± mild/moderate ischemia)	Plantar/non-plantar 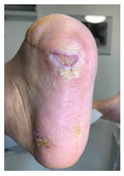	Lisfrank 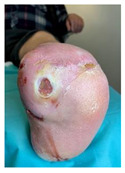 Chopard 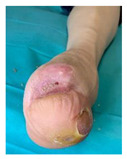	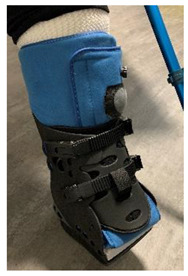
		Heel 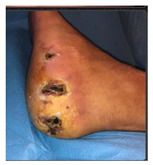		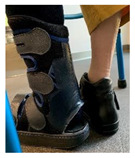 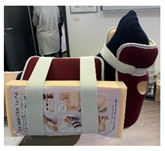
	Bilateral DFU/bilateral DFU + mild/moderate infection/± mild/moderate ischemia)	Plantar/non-plantar	Toes, transmetatarsals, rays, Lisfrank, Chopard	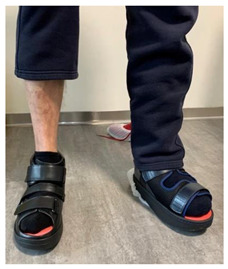

**Table 2 jcm-12-06516-t002:** Participants’ characteristics.

Participants’ Characteristics	Monitored Group (*n* = 45)
Age (years), arithmetic mean ± standard deviation	61.91 ± 9.31
Male, no. (%)	38 (84.4)
Female, no. (%)	7 (15.6)
BMI (kg/m^2^), median (25th–75th percentile)	29.4 (27.47; 31.96)
Shoe size (EU size conversion)	43 (42; 44)
Ex-smoker and smoker, no. (%)	11 (26.2)
Non-smokers, no. (%)	34 (73.8)
Active lifestyle, no. (%)	14 (31.1)
Rural, no. (%)	18 (40.9)
DM duration (years), arithmetic mean ± standard deviation	14.27 ± 8.05
HbA1c (%), median (25th–75th percentile)	7.3 (6.35; 8.75)
Diabetic peripheral neuropathy, no. (%)	
Without	0 (0.0)
Mild	1 (2.2)
Moderate	2 (4.4)
Severe	42 (93.3)
Vitamin B supplementation, no. (%)	4 (10.5)
With other neuropathy medication, no. (%)	4 (9.5)
Peripheral arterial disease, no. (%)	33 (73.3)
High blood pressure, no. (%)	32 (84.2)
Cardiovascular disease, no. (%)	19 (48.7)

no.—number; *n*—absolute frequency; BMI—body mass index; EU—European; DM—diabetes mellitus; HbA1c—glycated hemoglobin.

**Table 3 jcm-12-06516-t003:** Participants’ diabetic foot characteristics.

Participants’ Diabetic Foot Characteristics	Monitored Group (*n* = 45)
Number of present DFUs on podiatric assessment, no. (%)	
One	40 (88.9)
Two	4 (8.9)
Three	1 (2.2)
Actual DFU localization, no. (%)	
Left foot	22 (48.9)
Right foot	21 (46.7)
Both feet	2 (4.4)
Actual DFU duration at the inclusion (weeks), median (25th–75th percentile)	8 (3.5; 14)
DFU in the past, no. (%)	30 (66.7)
Off-loading for previous DFUs other than actual DFUs, no. (%)	2 (4.4)
No. of previous amputations, no. (%)	
One	17 (37.8)
Two	6 (13.3)
Three	3 (6.7)
Four	0 (0.0)
Five	1 (2.2)
Charcot foot, no. (%)	7 (15.6)
Foot deformities, no. (%)	26 (57.8)
Revascularization, no. (%)	
Without	21 (46.7)
Endovascular	20 (44.4)
Bypass	4 (8.9)

DFU—diabetic foot ulcer; no.—number; *n*—absolute frequency.

**Table 4 jcm-12-06516-t004:** Comparison between the groups with and without an RCW.

Parameters	Without Removable Walker (*n* = 17)	With Removable Walker (*n* = 28)	*p*
Age (years), arithmetic mean ± standard deviation	61.47 ± 8.91	62.18 ± 9.7	0.808 ^a^
Male, no. (%)	17 (100)	21 (75)	**0.034** ^b^
Female, no. (%)	0 (0)	7 (25)	
BMI (kg/m^2^), median (25th–75th percentile)	29.93 (27.04; 31.14)	28.99 (27.41; 31.98)	0.833 ^c^
Rural, no. (%)	11 (64.7)	7 (25.9)	**0.011** ^d^
Ex-smoker and smoker, no. (%)	5 (31.3)	6 (23.1)	0.720 ^d^
Non-smoker, no. (%)	12 (68.7)	22 (76.9)	
Lifestyle active, no. (%)	6 (35.3)	8 (28.6)	0.637 ^d^
HBP, no. (%)	9 (64.3)	23 (95.8)	**0.018** ^d^
CVD, no. (%)	7 (50)	12 (48)	1.000 ^d^
DM duration (years), arithmetic mean ± standard deviation	13.7 ± 7.11	14.68 ± 8.81	0.716 ^a^
HbA1c, median (25th–75th percentile)	7.65 (6.6; 8.55)	7.05 (6.35; 8.8)	0.830 ^c^
DPN			
Mild	1 (5.9)	0 (0)	0.396 ^c^
Moderate	1 (5.9)	1 (3.6)	
Severe	15 (88.2)	27 (96.4)	
Actual DFU duration			
≤4 weeks (1 month)	5 (29.4)	10 (35.7)	0.762 ^c^
5–8 weeks (2 months)	4 (23.5)	6 (21.4)	
9–16 weeks (3 -4 months)	4 (23.5)	5 (17.9)	
17–32 weeks (5–8 months)	2 (11.8)	1 (3.6)	
33–64 weeks (9–16 months)	2 (11.8)	1 (3.6)	
65–208 weeks (10–16 months)	0	3 (10.7)	
Not known	0	1 (3.6)	
No. of amputations in the past, arithmetic mean ± standard Deviation	0.65 ± 1.27	1.14 ± 0.93	**0.018** ^c^
No. of present DFUs/patient on podiatric assessment			
1	15 (88.2)	25 (89.3)	0.650 ^c^
2	2 (11.8)	2 (7.1)	
3	0 (0)	1 (3.6)	
Foot			
Left	4 (23.5)	18 (64.3)	**0.007** ^d^
Right	13 (76.5)	8 (28.6)	
Both	0 (0)	2 (7.1)	
PAD, no. (%)	7 (41.2)	20 (74.1)	**0.029** ^d^
Charcot foot, no. (%)	1 (5.9)	6 (21.4)	0.227 ^b^
Deformities, no. (%)	8 (47.1)	18 (64.3)	0.257 ^d^
DFUs in the past, no. (%)	9 (52.9)	21 (75)	0.128 ^d^
No. of pairs of protective footwear after initial podiatric evaluation			
0	0 (0)	1 (3.6)	0.710 ^c^
1	14 (82.4)	23 (82.1)	
2	3 (17.6)	4 (14.3)	
Lost from the study, no. (%)	5 (29.4)	4 (14.3)	0.265 ^b^
Healed, no. (%)	7 (58.3) ^1^	18 (75) ^2^	0.446 ^d^
Infection, no. (%)	3 (17.6)	6 (21.4)	1.000 ^b^
With re-ulceration, no. (%)	0 (0) ^3^	2 (11.1) ^4^	0.526 ^b^
With amputations during follow-up, no. (%)	1 (5.9)	1 (3.6)	1.000 ^b^
No. of weeks to complete healing			
≤4 weeks (1 month)	3 (42.9) ^3^	2 (11.1) ^4^	0.295 ^c^
5–8 weeks (2 months)	4 (57.1) ^3^	3 (16.7) ^4^	
9–16 weeks (3 -4 months)	0 (0) ^3^	9 (50) ^4^	
17–52 weeks (5–13 months)	0 (0) ^3^	4 (22.2) ^4^	
Total cost (EUR), arithmetic mean ± standard deviation	308.11 ± 87.29	425.77 ± 93.33	**<0.001** ^c^

no.—number; *n*—absolute frequency; BMI—body mass index; HBP—high blood pressure; CVD—cardiovascular disease; DM—diabetes mellitus; HbA1c—glycated hemoglobin; DPN—diabetic peripheral neuropathy; DFU—diabetic foot ulcer; PAD—peripheral arterial disease;^1^ from *n* = 12 (without patients who were lost from the study); ^2^ from *n* = 24 (without patients who were lost from the study); ^3^ from *n* = 7 (patients who were healed); ^4^ from *n* = 18 (patients who were healed); ^a^ *t*-test; ^b^ Fisher’s exact test; ^c^ Wilcoxon rank-sum test; ^d^ chi-square test.

**Table 5 jcm-12-06516-t005:** Survival Cox regression for the associated factors with the time to heal in remote patients treated at home during COVID-19.

	Univariate Analysis		Multivariate Analysis	
Parameters	*p*	HR (95% Confidence Interval)	*p*	HR (95% Confidence Interval)
Age (years)	**0.025**	1.05 (1.01; 1.09)	**0.026**	1.05 (1.01; 1.09)
Male	0.579			
BMI (kg/m^2^)	0.772			
Rural	0.316			
Ex-smoker and smoker	0.586			
Lifestyle active	0.851			
HBP	**0.047**	2.79 (1.01; 7.71)	**0.020**	3.44 (1.21; 9.76)
CVD	0.275			
DM duration (years)	0.440			
HbA1c	0.159			
DPN	0.451			
Number of amputations in the past	0.870			
Number of present DFUs/patient on podiatric assessment	0.134			
Foot left/right	0.666			
Foot one/both	**0.039**	4.93 (1.08; 22.42)	0.232	
PAD	0.156			
Charcot foot	0.379			
Deformities	0.547			
Revascularization (with/without)	0.902			
DFU in the past	0.625			
Number of pairs of protective footwear after initial podiatric evaluation	0.176			
Removable walker	0.884			
Infection	0.371			
Total cost (EUR)	0.404			

HR—hazard ratio; BMI—body mass index; HBP—high blood pressure; CVD—cardiovascular disease; DM—diabetes mellitus; HbA1c—glycated hemoglobin; DPN—diabetic peripheral neuropathy; DFU—diabetic foot ulcer; PAD—peripheral arterial disease.

## Data Availability

Not applicable.

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
