# Peer review of "Patients’ Buying Behavior for Non-Reimbursed Off-Loading Devices Used in Diabetic Foot Ulcer Treatment—An Observational Study during COVID-19 Pandemic from a Romanian Physical Therapy Unit"

_jcm, 2023, doi:10.3390/jcm12206516_

Round 1

Reviewer 1 Report

Dear authors,

Thank you for inviting me to review your manuscript, you have done an enormous amount of work that has been translated into an extensive manuscript. I would like to make some remarks in order to improve your work. 

-The introduction is extensive, although an important number of bibliographic references are more than 10 years old. Check if it is possible to change these references for more recent ones. Particularly important is reference 28, which refers to the discharge recommendations of the international diabetic foot working group. You refer to the 2020 recommendations, but recently (May 2023) the update has been published: Bus SA, Armstrong DG, Crews RT, Gooday C, Jarl G, Kirketerp-Moller K, Viswanathan V, Lazzarini PA. Guidelines on offloading foot ulcers in persons with diabetes (IWGDF 2023 update). Diabetes Metab Res Rev. 2023 May 25:e3647. doi: 10.1002/dmrr.3647. Epub ahead of print. PMID: 37226568. You should revise the text and adjust the manuscript in the aspects where the old recommendations have been followed.

-The method should explain the sampling system used, as well as the recruitment system.

You provide two very important variables in the study of diabetic foot, which are neuropathy and peripheral arterial disease, but they do not explain how the presence of neuropathy was determined or from what parameters the existence of PAD was determined (for example, existence of pulses, value of intima-mediated blood pressure, etc.).

No information is provided on the type of dressing used in each case. They were either moist wound dressings or gauze dressings. This is important because it is recognized that there were problems of tolerance or allergy to some of the dressings.

In the statistical analysis, it should be indicated whether a normality test was performed to determine the convenience of performing a parametric or non-parametric test.

In Table 2, it is not clear whether the percentage provided refers to smokers or ex-smokers. In Table 4, the p-values should appear with the same number of decimal places, and it should be indicated in the table footnote which test was used to obtain each p-value.

I would like to point out my opinion about the title. The title does not correspond to what is really presented in the work, since it does not explain the reasons why the subjects decide on one system of offloading versus another, it simply presents results associated with the offloading system used. In my opinion, this is simply a case series describing the offloading systems used in a series of patients with diabetic foot ulcer, although this is done in a very detailed and deep manner. Ideally, it would have been ideal to include variables to evaluate this.  Include for example a variable to measure satisfaction with the offloading system or a variable for the subject to explain why they chose one system over another. As you state the cost does not seem to be a feasible explanation.

Finally, one aspect that I find lacking is that no information is provided on the family or community support that these subjects had, which is important in performing self-care for diabetic foot ulcers and may influence the effectiveness of one system versus another or the appearance of the different situations during remote monitoring.

I hope that my comments will help you to improve this manuscript.  

Good work

Author Response

Answers to comments addressed by the Reviewer 1

Reviewer comment

Author response

1.

The introduction is extensive, although an important number of bibliographic references are more than 10 years old. Check if it is possible to change these references for more recent ones. Particularly important is reference 28, which refers to the discharge recommendations of the international diabetic foot working group. You refer to the 2020 recommendations, but recently (May 2023) the update has been published: Bus SA, Armstrong DG, Crews RT, Gooday C, Jarl G, Kirketerp-Moller K, Viswanathan V, Lazzarini PA. Guidelines on offloading foot ulcers in persons with diabetes (IWGDF 2023 update). Diabetes Metab Res Rev. 2023 May 25:e3647. doi: 10.1002/dmrr.3647. Epub ahead of print. PMID: 37226568.

You should revise the text and adjust the manuscript in the aspects where the old recommendations have been followed.

Thank you very much for your appreciations and pertinent comments.

We tried to improve the Introduction section.

Where we found more recent references, we have revised references older than 10 years.

We have changed in the “Introduction” chapter the 2020 IWGDF reference (ref 28) with the recently published one (Bus SA, Armstrong DG, Crews RT, Gooday C, Jarl G, Kirketerp-Moller K, Viswanathan V, Lazzarini PA. Guidelines on offloading foot ulcers in persons with diabetes (IWGDF 2023 update). Diabetes Metab Res Rev. 2023 May 25:e3647. doi: 10.1002/dmrr.3647. Epub ahead of print. PMID: 37226568.) updates, as per your pertinent observation.

Our monitoring period started in March 2020 and ended at February 2022. We have used and followed the recommendations/classifications according to the 2020 IWGDF guidelines, thereafter in the manuscript text we are not able to modify the 2020 IWGDF guidelines with the latest version (2023).

2.

The method should explain the sampling system used, as well as the recruitment system

We added:

All patients who were addressed for off-loading procedures and met the inclusion criteria, were consecutively included in the study..

3.

You provide two very important variables in the study of diabetic foot, which are neuropathy and peripheral arterial disease, but they do not explain how the presence of neuropathy was determined or from what parameters the existence of PAD was determined (for example, existence of pulses, value of intima-mediated blood pressure, etc.).

The two variables used, PAD and DPN were taken from patient’s medical files and were previously diagnosed by the medical team (vascular surgeons’ assessment and measurements), all being done before the patients were sent for the off-loading treatment session. PAD was diagnosed using ankle and brachial index values (according to the (according to the Intersocietal PAD guideline (2023 update - https://iwgdfguidelines.org/wp-content/uploads/2023/07/IWGDF-2023-05-PAD-Guideline.pdf, while

DPN was diagnosed using Michigan Neuropathy Screening Instrument (MNSI), Neuropathy Disability Score (NDS), The Neurological Symptom Score (NSS), while DPN severity was diagnosed using the values in mV for vibration perception threshold (VPT) measured using the Neurothesiometer.

4.

No information is provided on the type of dressing used in each case. They were either moist wound dressings or gauze dressings. This is important because it is recognized that there were problems of tolerance or allergy to some of the dressings.

The dressings used were chosen by the medical team (vascular surgeon before patients were sent for off-loading). In case of wounds with excessive exudate, moist wound dressings were used with high moist absorption properties (Zetuvit-Hartman, Sorbalgon-Hartman), while for less moist or dry wounds, gaze dressings were used (Medicomp, Hartman).

Allergies were seen in case patients didn’t t respect the dressings indications and have used other available non-specific dressings. One particular allergy has been seen in the case of using the padding material directly on the skin due to the unavailability of cotton tubular bandage that protects the direct contact of wool padding with the skin. Another particular irritation was seen in a patient that used gaze dressing with adherent adhesive on the skin and not semi elastic gaze bandage as per the medical team recommendations. No particular allergies were directly related to the wound dressings. The allergies were mainly due to adhesive, or wool padding material.

5.

In the statistical analysis, it should be indicated whether a normality test was performed to determine the convenience of performing a parametric or non-parametric test.

We added the requested in the manuscript: “The qualitative characteristics of the two subgroups were compared using statistical tests: Chi-square test/Fisher’s exact test. After using Shapiro-Wilk test for normal distribution, the quantitative characteristics were compared with t test or with Wilcoxon rank-sum test.”

6.

In Table 2, it is not clear whether the percentage provided refers to smokers or ex-smokers.

In Table 4, the p-values should appear with the same number of decimal places, and it should be indicated in the table footnote which test was used to obtain each p-value.

The percentage provided refers to both smoker and ex-smokers. Data were collected from medical files where we only found two categories: smokers (which included also ex-smokers) and non-smokers.

We added the non-smokers no hoping now it is clearer.

The number of decimal places has been revised in Table 4.

The Test for p value calculation has been added in the footnote and marked in the Table 4. 

7.

The title does not correspond to what is really presented in the work, since it does not explain the reasons why the subjects decide on one system of offloading versus another, it simply presents results associated with the offloading system used. In my opinion, this is simply a case series describing the offloading systems used in a series of patients with diabetic foot ulcer, although this is done in a very detailed and deep manner. Ideally, it would have been ideal to include variables to evaluate this.  Include for example a variable to measure satisfaction with the offloading system or a variable for the subject to explain why they chose one system over another. As you state the cost does not seem to be a feasible explanation.

We modified the title “Factors influencing patients’ acceptance and preferences for non-reimbursed off-loading devices used in the treatment of diabetic foot ulcer. A pandemic observational study from a Physical Therapy Unit” in:

“Patients buying behavior for non-reimbursed off-loading devices used in the diabetic foot ulcer treatment. An observational study during Covid-19 pandemic from a Romanian Physical Therapy Unit”, in accordance with your pertinent remarks.

The suggested variables were not used at the first off-loading session, neither at the end of the monitoring period. 

We added as a limitation of our study not using questionnaires/variables able to explain why they chose one system over another.

By changing the confounding article “Title” we hope these aspects were resolved.

8.

Finally, one aspect that I find lacking is that no information is provided on the family or community support that these subjects had, which is important in performing self-care for diabetic foot ulcers and may influence the effectiveness of one system versus another or the appearance of the different situations during remote monitoring.

The support that these patients had for self-foot care was provided using:

-written information released by the medical team on the dressing procedure.

-written/video captures of the manual instructions for all off-loading devices used

-video /photography-based assistance from the main researcher (physiotherapist) on patients request, or in case during remote monitoring any different situations appeared.

Family support was the most common situation encountered in our study group, but some patients living alone benefit only from self-wound care under the medical team instructions.

Difficult cases, such as patients unable to self-care their foot, had the help of a nurse in their district was requested.

No organized community support was available for any of the patients in the study group.

Reviewer 2 Report

The research presents a miscellany of a how authors manage from an offloading vision all their patients in an outpatient setting.

After revising the title, I think the authors did not meet the expectations of the suggested title. Patients acceptance and preferences are not described in the manuscript. Please revise further papers about the topic under consideration:

-van Netten JJ, Jannink MJ, Hijmans JM, Geertzen JH, Postema K. Use and usability of custom-made orthopedic shoes. J Rehabil Res Dev. 2010;47(1):73-81.

-Knowles EA, Boulton AJ. Do people with diabetes wear their prescribed footwear? Diabet Med. 1996;13(12):1064-8.

-van Netten JJ, Dijkstra PU, Geertzen JH, Postema K. What influences a patient’s decision to use custom-made orthopaedic shoes? BMC Musculoskelet Disord. 2012;13:92.

-van Netten JJ, Jannink MJ, Hijmans JM, Geertzen JH, Postema K. Patients’ expectations and actual use of custom-made orthopaedic shoes. Clin Rehabil. 2010;24(10):919-27.

-López-Moral M, Molines-Barroso RJ, Herrera-Casamayor M, García-Madrid M, García-Morales E, Lázaro-Martínez JL. Usability of Different Methods to Assess and Improve Adherence to Therapeutic Footwear in Persons with the Diabetic Foot in Remission. A Systematic Review. Int J Low Extrem Wounds. 2023 Aug 7:15347346231190680. 

It looks like a protocol of their unit more than a research paper.

The results doesn´t meet the objective of the paper neither the title.

Author Response

Thank you for your pertinent comments.

Answers to comments addressed by the Reviewer 2

Reviewer comment

Author response

1

After revising the title, I think the authors did not meet the expectations of the suggested title. Patients acceptance and preferences are not described in the manuscript. Please revise further papers about the topic under consideration:

-van Netten JJ, Jannink MJ, Hijmans JM, Geertzen JH, Postema K. Use and usability of custom-made orthopedic shoes. J Rehabil Res Dev. 2010;47(1):73-81.

-Knowles EA, Boulton AJ. Do people with diabetes wear their prescribed footwear? Diabet Med. 1996;13(12):1064-8.

-van Netten JJ, Dijkstra PU, Geertzen JH, Postema K. What influences a patient’s decision to use custom-made orthopaedic shoes? BMC Musculoskelet Disord. 2012;13:92.

-van Netten JJ, Jannink MJ, Hijmans JM, Geertzen JH, Postema K. Patients’ expectations and actual use of custom-made orthopaedic shoes. Clin Rehabil. 2010;24(10):919-27.

-López-Moral M, Molines-Barroso RJ, Herrera-Casamayor M, García-Madrid M, García-Morales E, Lázaro-Martínez JL. Usability of Different Methods to Assess and Improve Adherence to Therapeutic Footwear in Persons with the Diabetic Foot in Remission. A Systematic Review. Int J Low Extrem Wounds. 2023 Aug 7:15347346231190680.

We agree with this pertinent remark, and concluded with the Title changing according with your comments and suggested references as follows:

“Patients buying behavior for non-reimbursed off-loading devices used in the diabetic foot ulcer treatment. An observational study during Covid-19 pandemic from a Romanian Physical Therapy Unit”.

We hope this Title modification better explains the study main aim.

2.

It looks like a protocol of their unit more than a research paper.

We revised the aim of the study according to your pertinent observations regarding the article title.

We have analyzed how the patient’s buying behavior impacted on the healing results.

In our opinion this paper is research.

Not in all countries the DFU treatment is supported by the National Health Insurance Company, so patients are buying devices, and the buying behavior and its consequences was the main objective studied in this paper. 

Table 4 and 5 present the analytical research.

3.

The results doesn´t meet the objective of the paper neither the title.

The aim of the study was revised, due to some words miss-understanding/miss-use.

 Title was revised as well, according to the pertinent comments you addressed us. We now hope that the results meet the objective of the paper.

Reviewer 3 Report

This is a well-analyzed paper.

Introduction

Briefly mention the prevalence or risk factors that were frequently mentioned in the introduction and include them in the discussion part. The content is too lengthy to be covered in the introduction. And mention your foot care program more.

Authors classified PROM by year, country and institution, journal and author, and keywords.

Materials and Methods

Well organized.

It would be a good idea to organize the photos to a size that makes them easier to see.

Results

Please mark them as male and female in table 2,4

Discussion

It covers too much content, making it difficult for readers to read. Please keep it more concise by only covering what is essential to the topic.

Author Response

Thank you for your pertinent comments.

Answers to comments addressed by the Reviewer 3

Reviewer comment

Author response

1

Introduction:

Briefly mention the prevalence or risk factors that were frequently mentioned in the introduction and include them in the discussion part.

The content is too lengthy to be covered in the introduction.

And mention your foot care program more.

Authors classified PROM by year, country and institution, journal and author, and keywords.

We add the suggested remarks on the prevalence of risk factors in the “Discussion” chapter.

 We are sorry, but we do not understand your request. Which foot care program are you referring at?

We did not use any PROM in this study.

2.

Materials and Methods

Well organized.

It would be a good idea to organize the photos to a size that makes them easier to see.

The pictures will respect the Journal requests. We will take care that the pictures will be clear and big enough to be well and easier seen.

3.

Results

Please mark them as male and female in table 2,4.

The requested aspects have been marked in Tables 2,4 as per your remarks with male and female.

4.

Discussion

It covers too much content, making it difficult for readers to read. Please keep it more concise by only covering what is essential to the topic.

We have reduced the length and better organized the Discussions content as per your remarks.
